

# Shifts in dominance of benthic communities along a gradient of water temperature and turbidity in tropical coastal ecosystems

Ludi Parwadani Aji[1,2,3], Diede Louise Maas[1], Agustin Capriati[1], Awaludinnoer Ahmad[4], Christiaan de Leeuw[1] and Leontine Elisabeth Becking[1,2]

[1] Wageningen University and Research, Wageningen, The Netherlands
[2] Naturalis Biodiversity Center, Leiden, The Netherlands
[3] Research Center for Oceanography, National Research and Innovation Agency, Jakarta, Indonesia
[4] Yayasan Konservasi Alam Nusantara, Sorong, Southwest Papua, Indonesia

Corresponding authors
Ludi Parwadani Aji, ludi.aji@wur.nl
Leontine Elisabeth Becking, lisa.becking@wur.nl

## ABSTRACT

Tropical coastal benthic communities will change in species composition and relative dominance due to global (*e.g.*, increasing water temperature) and local (*e.g.*, increasing terrestrial influence due to land-based activity) stressors. This study aimed to gain insight into possible trajectories of coastal benthic assemblages in Raja Ampat, Indonesia, by studying coral reefs at varying distances from human activities and marine lakes with high turbidity in three temperature categories (<31 °C, 31–32 °C, and >32 °C). The benthic community diversity and relative coverage of major benthic groups were quantified via replicate photo transects. The composition of benthic assemblages varied significantly among the reef and marine lake habitats. The marine lakes <31 °C contained hard coral, crustose coralline algae (CCA), and turf algae with coverages similar to those found in the coral reefs (17.4–18.8% hard coral, 3.5–26.3% CCA, and 15–15.5% turf algae, respectively), while the higher temperature marine lakes (31–32 °C and >32 °C) did not harbor hard coral or CCA. Benthic composition in the reefs was significantly influenced by geographic distance among sites but not by human activity or depth. Benthic composition in the marine lakes appeared to be structured by temperature, salinity, and degree of connection to the adjacent sea. Our results suggest that beyond a certain temperature (>31 °C), benthic communities shift away from coral dominance, but new outcomes of assemblages can be highly distinct, with a possible varied dominance of macroalgae, benthic cyanobacterial mats, or filter feeders such as bivalves and tubeworms. This study illustrates the possible use of marine lake model systems to gain insight into shifts in the benthic community structure of tropical coastal ecosystems if hard corals are no longer dominant.

## INTRODUCTION

Tropical coastal ecosystems are changing rapidly due to global stressors caused by climate change, predominantly in the form of increasing water temperatures (*Hughes et al., 2018*;

*Obura et al., 2022*), as well as local anthropogenic activities that result in local pollution, such as sedimentation and eutrophication (*e.g.*, *Duprey, Yasuhara & Baker, 2016*; *Otaño Cruz et al., 2019*; *Lesser, 2021*). Coral reefs are among the world's most productive and biodiverse ecosystems that support the livelihood of millions of people in tropical and sub-tropical areas (*Eddy et al., 2021*; *Mellin et al., 2022*). It is therefore important to understand how these ecosystems are changing under both increasing seawater temperatures and local stressors. Eutrophication can decrease oxygen concentration and water quality, and reduced algal grazing because of overfishing can disrupt the balance of the reef ecosystem (*Smith et al., 2003*; *Fabricius, 2011*; *Kennedy et al., 2013*; *Lesser, 2021*). Coastal development contributes to sedimentation and pollution, leading to high water turbidity levels, further exacerbating coral reef deterioration (*Fabricius, 2005*; *Pandolfi et al., 2005*; *Otaño Cruz et al., 2019*). The combination of global and local stressors can negatively impact the coverage and dominance of corals while enhancing the growth of other benthic groups that can colonize available substrate and thrive in the new environmental conditions (*de Bakker et al., 2016*; *de Bakker et al., 2017*; *Pawlik & McMurray, 2020*; *Tebbett, Connolly & Bellwood, 2023a*).

Environmental stressors could cause a benthic community shift from coral-dominated reefs towards reefs dominated by other emergent non-reef building groups (*Tebbett et al., 2023b*). Most documented examples of shifts in community composition in the past two decades are from coral-dominated to macroalgae-dominated or turf algae-dominated systems (*Pandolfi et al., 2005*; *Tebbett & Bellwood, 2019*; *Crisp, Tebbett & Bellwood, 2022*; *Rivas et al., 2023*). In recent years, there have been reports of other benthic groups, such as benthic cyanobacterial mats (BCMs), sponges, and soft corals, becoming emergent players in the reefs. For example, BCMs have become more prevalent in reefs in the Caribbean (*Brocke et al., 2015*; *de Bakker et al., 2017*) and in some locations in the Indo-Pacific (*Ford et al., 2018*; *Ford et al., 2021*). BCMs can be toxic (*Nagle & Paul, 1998*), cause local anoxia (*Brocke et al., 2015*), and prevent coral larvae and other benthic organisms from settling due to the production of harmful compounds and their disruptive impact on the surrounding marine environment (*Kuffner et al., 2006*; *Ford et al., 2018*; *Reverter et al., 2020*). Sponges are predicted to become more dominant in tropical reefs due to their resilience to rising sea temperatures, as well as their potential to outcompete certain coral species under changing environmental conditions (*Bell et al., 2013*; *Bell et al., 2018*; *de Bakker et al., 2017*; *Pawlik & McMurray, 2020*). There has been a possible shift noticed to soft coral dominance in benthic communities in tropical coastal ecosystems in the Indo-Pacific because soft coral may be more resistant to warming temperatures and nutrient enrichment than hard coral (*Baum et al., 2016*; *Mezger et al., 2022*; *Reverter et al., 2022*). Shifts from coral-dominated reefs to reefs dominated by other taxonomic groups can negatively affect the coral reef ecosystem (*Hughes et al., 2010*; *Tebbett, Connolly & Bellwood, 2023a*), since these shifts can lead to decreases in structural complexity (*Alvarez-filip et al., 2009*), and a reduction in both habitat availability for various reef species (*Graham & Nash, 2013*) and ecological functions within the food webs (*Ford et al., 2018*; *Nelson, Kelly & Haas, 2022*; *Mortimer et al., 2023*). The function of coastal tropical ecosystems could change significantly based on these potential shifts in dominance of benthic groups.

The Coral Triangle area is particularly suited to studying the impacts of stressors on benthic community diversity and structure as it is the global hotspot for marine biodiversity (*Hoeksema, 2007*; *Veron et al., 2011*). The Raja Ampat islands of Southwest Papua, Indonesia, located in the center of the Coral Triangle area, are home to about 553 scleractinian coral species (*Veron et al., 2011*), with coral coverage ranging from 25–50% (*Hadi et al., 2019*; *Purwanto et al., 2021*). However, like other reef ecosystems worldwide, the Raja Ampat waters are facing challenges due to environmental changes associated with human activity and climate change (*Mangubhai et al., 2012*; *King, 2017*; *Ampou et al., 2020*). The number of tourists to the Raja Ampat islands continues to rise, which may negatively impact the coastal ecosystem (*King, 2017*; *Papilaya, Boli & Nikijuluw, 2019*; *Maas et al., 2020*; *Purwanto et al., 2021*). Coral bleaching and disease have also been observed because of environmental stresses such as seawater temperature fluctuation, decreasing salinity, high sedimentation, and turbidity (*Ampou et al., 2020*; *Johan et al., 2020*; *Subhan et al., 2020*). Identifying the drivers of benthic community diversity in marine habitats is crucial for gaining deeper insights into how the benthic communities may shift.

Most relevant research to date has focused on the ecological responses of reefs (time scales of several decades and under ambient temperature conditions (*de Bakker et al., 2017*; *Giorgi et al., 2022*; *Caballero-aragón et al., 2023*) or of selected species within an experimental set up of a coral reef community to increased temperatures over short time scales (weeks/months; *Ullah et al., 2018*; *Grottoli et al., 2021*; *Mezger et al., 2022*), yet little is known about the long-term response of whole communities to changes in temperature and environmental conditions. Marine lakes with high water temperatures and high turbidity may provide insights into potential shifts in dominance of benthic groups in tropical coastal ecosystems (*Hamner & Hamner, 1998*; *Becking et al., 2011*; *Maas et al., 2023*). Marine lakes are anchialine systems, which are small bodies of landlocked seawater with marine characteristics through subterranean connections to the sea through tunnels, fissures, or pores in the surrounding karstic limestone (*Holthuis, 1973*; *Hamner & Hamner, 1998*). The biota found in marine lakes originate from the sea as larval propagules enter the lakes through these subterranean connections. Both environmental parameters and degree of connection seem to influence genetic and species diversity in marine lakes (*Becking et al., 2016*; *Rapacciuolo et al., 2019*; *Cleary & Polónia, 2020*; *Aji et al., 2023*; *Maas et al., 2023*). The degree of connection between marine lakes and the sea varies, with the higher the connection, the more similar their abiotic and biotic conditions due to increased inflow of seawater into the marine lakes (*Becking et al., 2011*; *Meyerhof et al., 2016*). Highly-connected marine lakes tend to have lower water retention and temperatures because of dilution through seawater turnover, while marine lakes with low connectivity experience higher water retention, and may have higher terrestrial nutrient input (runoff) from surrounding tropical rainforests. Marine lakes provide natural states of predicted environmental scenarios, including increased water temperatures and land-based turbidity. Hence, marine lakes provide a system to understand the interactions of major benthic groups under coastal conditions of high temperature and turbidity.

We studied 23 open water reef sites and 11 marine lakes with three temperature categories (<31 °C, 31–32 °C, and >32 °C) in Raja Ampat, Southwest-Papua, Indonesia. The diversity
and composition of benthic communities were compared among coral reefs and marine lakes in Raja Ampat, Indonesia to give insight into which benthic groups emerge when corals are no longer dominant. The objectives of this study were to: (i) quantify the variation in benthic community diversity across different coral reef sites and marine lakes in three temperature categories, and; (ii) identify the relative contributions of habitat, environmental variables, geographic location, human activity, and depth to the variation in benthic assemblages.

## MATERIALS AND METHODS

### Study sites and marine lake profiling

This study was conducted in Raja Ampat, Southwest Papua, Indonesia (Fig. 1). Field works were approved by *Kementrian Riset, Teknologi, dan Pendidikan Tinggi* (1353/FRP/E5/Dit.KI/III/2018), and *Lembaga Ilmu Pengetahuan Indonesia* (B-7/IPK.2/KS.01/I/2020). In total, 23 reef sites were surveyed from two regions of Raja Ampat between April and May 2018. Specifically, six sites were surveyed in Dampier Strait and 17 sites in Misool, at both 5m and 10m depths at all 23 sites (Fig. 1). Seawater temperature and salinity was relatively stable across the period 2016–2020, approximating 30 °C and 30 ppt, respectively. In the Misool area, 11 marine lakes were surveyed from January to February 2020 and classified into three temperature categories: <31 °C (two marine lakes), 31–32 °C (four marine lakes), and >32 °C (five marine lakes; Fig. 1). The temperature categories are split based on a 1 °C increase from <31 °C to 31–32 °C and >32 °C. According to the Intergovernmental Panel on Climate Change (IPCC) scenario, the sea surface temperature might rise by 2.58 °C by the year 2100 (*IPCC, 2014*). Therefore, temperatures beyond 32 °C can be considered extreme for future coastal conditions. A coding system consistent with *Aji et al. (2023)* was used for the names of the marine lakes.

In the marine lakes, a YSI Professional Plus handheld multi-parameter was used to record abiotic environmental factors, including temperature (°C) and salinity (‰), at 1 m intervals starting one meter below the surface to a depth of 5 m. The water quality measurements were done in the afternoon. The maximum depth of each marine lake was measured using a handheld sonar system (Hawkeye), and the surface area (m²) was approximated using Google Earth Pro. To compare tidal fluctuation inside and outside the marine lake, HOBO water level loggers were deployed for at least 24 h in the marine lake and in the sea as a proxy for the degree of connection. The calculation of the fraction of tidal amplitude of the marine lake compared with the sea followed the methods outlined by *Maas et al. (2018)*. The degree of connection ranged from zero to one. The perimeter of the marine lakes consisted mainly of karstic rock and the marine lake floors were covered by fine sediments such as mud, plant litter from the surrounding forest, dead shells, and rock.

### Benthic surveys and analysis of photo quadrats

Benthic communities from the marine lakes and reefs were assessed using a photo quadrat transect method (*Hill & Wilkinson, 2004*). The transects were laid out at random, and subsequently the photos were photographed using an Olympus tough TG-6 digital camera

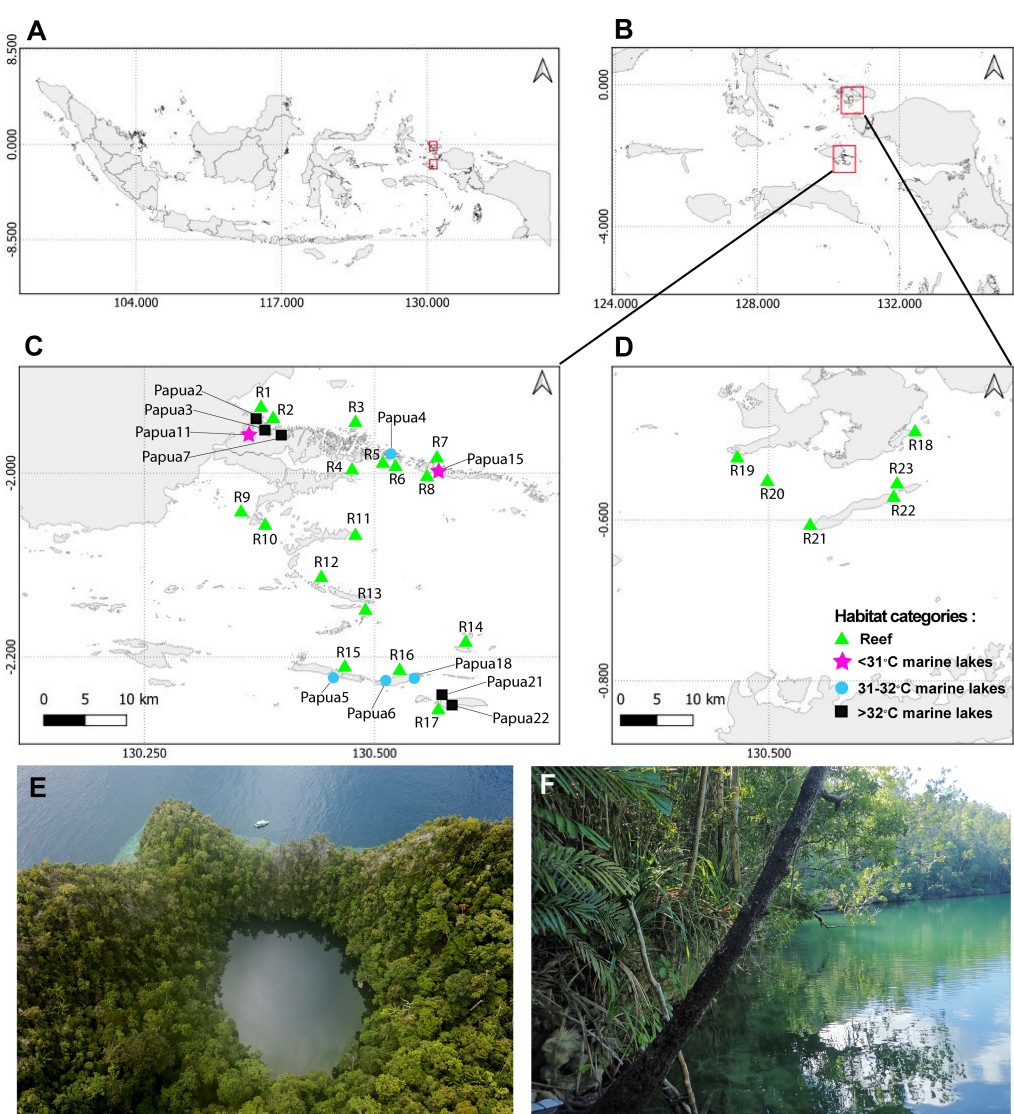

**Figure 1** **Sampling sites from 23 reefs and 11 marine lakes.** (A) Indonesia. (B) Overview of two geographic regions survey location in Raja Ampat, Southwest Papua. (C) Zoomed-in overview of Misool showing the locations of 17 of the reefs and the 11 marine lakes. (D) Zoomed-in overview of Dampier Strait showing the location of six of the reefs sampled in this study. (E) Aerial view of marine lake Papua 6 (photo credit: Christiaan de Leeuw) and (F) ground-level view of marine lake Papua 2 (photo credit: Ludi P. Aji).

along the transect line continuously from the starting point until the end of the transect line. In each of the coral reefs, three replicates consisting of three 15 m transect lines were placed parallel to the coastline by SCUBA diving at 5 m and 10 m depths. Photo quadrats of 25 × 25 cm were taken with 30 pictures per transect in the reef sites. The distance of human activity—such as a homestay, resort, village, aquaculture activity, or tourism area—to the reef sites was also determined as a proxy of anthropogenic pressure. Within each marine lake, three 15 m transect lines were positioned parallel to the wall of the marine lake,

approximately one meter below the lowest tide level. Each marine lake transect had a total of 60 pictures taken with a frame size of 25 × 15 cm. The total surface area observed per site was 6.75 m$^2$ in each marine lake and 5.625 m$^2$ in each coral reef. A statistical test was conducted to verify that coverage estimates from benthic communities within the marine lake from both frame sizes (25 × 25 cm *versus* 25 × 15 cm) did not differ significantly. A pairwise comparison between frame sizes within marine lake Papua 4 showed no significant differences in coverage estimates (Welch two paired sample *t*-test, $t = 0$, df 129.47, $p = 1$).

The frame photos were analyzed using the Coral Point Count with Excel extensions (CPCe) program to estimate the coverage of the different benthic organisms (*Kohler & Gill, 2006*). A uniform grid with 25 points per photo-quadrat was used and the benthic groups underneath each of the points were identified using a two-step grouping system with major (coarse functional group) and finer-scale categories of benthic groups (Table S1). The finer-scale benthic groups (66 in total) were classified into ten major groups: hard corals (14 groups), soft corals (3 groups), crustose coralline algae (1 group), turf algae (2 groups), sponges (8 groups), macroalgae (6 groups), benthic cyanobacterial mats (3 groups), bivalves (4 groups), other invertebrates (17 groups), and substrates (8 groups). The finer-scale benthic groups of hard corals and sponges were based on their growth form, whereas other live benthic groups were based on their taxa at the genus or family level. The presence/absence and abundance of benthic groups were identified through an analysis of the photos. The CPCe files were then combined into one dataset for analysis.

## Data analysis

All analyses were performed in R v.4.0.2 (*R Core Team, 2022*). For the environmental variables, a Pearson correlation analysis was employed to test for collinearity between the degree of connection, temperature, and salinity of the marine lakes.

The total number of live finer-scale benthic groups in each site was then used to calculate group richness. The Shannon diversity index was calculated on the abundance data per finer-scale benthic group with the *diversity* function in the *vegan* package (*Oksanen et al., 2019*). The Shannon diversity index considers group richness and evenness, with a high value indicating a high diversity and a lack of dominance by certain groups (*Shannon & Weaver, 1949*). This was followed by testing differences in Shannon diversity by habitat category using one-way non-parametric Kruskal–Wallis and then post-hoc Dunn test pairwise comparisons between habitats. In addition, Pearson correlation tests were performed between Shannon diversity and degree of connection and surface area of marine lakes.

For analyzing benthic community structure, data normality was tested before analysis using the *shapiro.test* function and visualized using the *hist* function in R. The homogeneity of variances was checked using the *levene test*. As the assumption of normality and homogeneity were violated for the raw data, square root transformation was used to normalize the data and correct for the influence of highly abundant benthic groups. Then, the *prob.table* function in R was used to calculate the proportions of benthic composition. The difference in proportions of each benthic group was tested using the non-parametric Kruskal–Wallis test followed by pairwise comparisons with a post-hoc Dunn test to determine which habitats differed significantly. Stacked bar graphs and boxplots were used
to visualize the coverage of the 10 major benthic groups per site and habitat category, respectively.

A permutational multivariate analysis with 999 permutations (PERMANOVA; *Anderson, 2001*) was performed using the *adonis* function (*vegan* package v.2.6.2) to test significant differences in community composition among habitat categories (*Oksanen et al., 2019*). The composition of benthic communities among habitats was visualized using non-metric multidimensional scaling (NMDS; *Legendre & Gallagher, 2001*). NMDS is an unconstrained ordination method used to find compositional variation and to relate this variation to the observed environmental variation (*Kenkel & Orlóci, 1986*). The *metaMDS* function was implemented with the Bray–Curtis dissimilarity matrix as input from benthic coverage data. Bray–Curtis distance, a measure to identify differences among groups of samples (*Bray & Curtis, 1957*), was constructed using the *vegdist* function in the *vegan* package (*Oksanen et al., 2019*). The stress value and Shepard diagram were assessed to determine the appropriateness of the NMDS. An *enfvit* analysis was then performed to analyze the effect of the explanatory variables (marine lakes: degree of connection (serving as proxy for temperature and turbidity), and geographic distance; reefs: geographic distance, human activity, and depth) on the benthic community assemblage across sites. The function *envfit* from the vegan package was used to fit explanatory variable vectors and finer-scale benthic group factors on the NMDS to assess the significance of these variables.

To identify benthic organisms that contributed most to the composition of the habitat (indicator species), the *indicspecies* package v.1.7.12 in R was used on the finer-scale benthic group coverage data across habitats (*De Cáceres & Legendre, 2009*). The *multipatt* function and permutational *p*-value were generated to determine finer-scale benthic groups associated with each habitat. The effects of geographic distance on community composition were then tested using the Mantel test (*Mantel, 1967*) with the *mantel* function from the *vegan* R package (*Oksanen et al., 2019*). The geographic distance variable referred to the distance between survey sites and was calculated as the minimum pairwise distance in meters between coordinates from the center of each site using the *distm* function from the *geosphere* R package (*Hijmans et al., 2022*). Mantel tests were conducted on marine lake and reef sites across Misool area, only marine lakes, and only reef sites. To check for correlation between distance matrices, the Mantel test was performed using the *vegan* package with 999 permutations. Since the data were not normally distributed and variances were not homogeneous, Spearman correlations were performed.

## RESULTS

### Environmental conditions and characteristics
The environmental conditions and characteristics of the marine lakes and reefs included in this study are presented in Table 1. Temperatures ranged from approximately 30 °C to 38 °C, and salinity ranged from approximately 15 ppt to 30 ppt. Higher variations in temperature and salinity were found in >32 °C marine lake Papua 21 (standard deviation 1.04 °C and 2.14 ppt) and Papua 22 (standard deviation 3.01 °C and 3.53 ppt) than in any other marine lakes. The surface area of the marine lakes ranged between 3,700 m$^2$

**Table 1 Characterization and environmental parameters (mean ± standard deviation based on three sites per marine lake from 1–5 m) in marine lakes in Raja Ampat, Indonesia.** Connection to the sea is the ratio of the maximum tidal fluctuation in a marine lake divided by that of the adjacent open sea. Codes for the marine lakes based on *Aji et al. (2023)*.

| Code | Habitat | Temperature (°C) | Salinity (ppt) | Connection to the sea | Surface area (m$^2$) | Maximum depth (m) |
|------|---------|------------------|----------------|----------------------|---------------------|-------------------|
| Papua 15 | <31 °C | 30.5 ± 0.04 | 30.2 ± 0.13 | 0.9 | 10,300 | 34 |
| Papua 11 | <31 °C | 30.7 ± 0.07 | 28.4 ± 0.45 | 0.8 | 27,300 | 9 |
| Papua 5 | 31–32 °C | 31.3 ± 0.32 | 29.1 ± 0.30 | 0.3 | 3,700 | 5 |
| Papua 18 | 31–32 °C | 31.6 ± 0.18 | 28.7 ± 0.32 | 0.8 | 7,000 | 5 |
| Papua 4 | 31–32 °C | 31.7 ± 0.36 | 26.0 ± 0.82 | 0.8 | 13,750 | 20 |
| Papua 6 | 31–32 °C | 31.8 ± 0.26 | 28.3 ± 0.27 | 0.8 | 2,950 | 12 |
| Papua 3 | >32 °C | 32.4 ± 0.40 | 27.4 ± 0.98 | 0.5 | 20,800 | 8 |
| Papua 2 | >32 °C | 33.5 ± 0.35 | 25.3 ± 0.40 | 0.2 | 12,200 | 7 |
| Papua 21 | >32 °C | 34.9 ± 1.04 | 23.9 ± 2.14 | 0.1 | 18,950 | 13 |
| Papua 7 | >32 °C | 35.4 ± 0.15 | 15.3 ± 0.61 | 0.2 | 9,700 | 8 |
| Papua 22 | >32 °C | 37.6 ± 3.01 | 18.5 ± 3.53 | 0.1 | 23,100 | 12 |

and 27,300 m$^2$. Pearson correlation tests showed evidence of collinearity in marine lakes between the degree of connection and temperature ($r = -0.82$, $p = 0.002$) and salinity ($r = 0.69$, $p = 0.019$), as well as between temperature and salinity ($r = -0.89$, $p = 0.0002$).

## Diversity and coverage of benthic groups among habitats

Shannon diversity index and finer-scale group richness significantly differed among the habitats (Shannon diversity: 2.27 ± 0.21 *versus* 1.47 ± 0.38; and richness: 26 ± 3.8 *versus* 16 ± 6.8, Kruskal–Wallis: $p < 0.001$). Shannon index (Fig. 2) was significantly different among Reef 10 m and >32 °C marine lakes (Dunn-test: $p = 0.009$) , Reef 10 m and 31–32 °C marine lakes ($p = 0.032$), Reef 5 m and >32 °C marine lakes ($p = 0.013$), and Reef 5 m and 31–32 °C marine lakes ($p = 0.007$). Benthic group richness (Fig. S1) was also significantly different among Reef 10 m and >32 °C marine lakes ($p = 0.017$), Reef 10 m and 31–32 °C marine lakes ($p = 0.022$), Reef 5 m and >32 °C marine lakes ($p = 0.012$) and Reef 5 m and 31–32 °C marine lakes ($p = 0.017$). There was a significant Pearson correlation between the Shannon diversity index and degree of connection ($r = 0.66$, $p = 0.028$), but not between the Shannon diversity index and surface area ($r = 0.41$, $p = 0.21$; Fig. S2).

Major benthic group coverage varied across habitats (Fig. 3, Fig. S3, Table S2). Hard and soft coral was only observed in Reef 10 m (mean ± SD: 17.6% ± 6.1% and 6.6% ± 3.9%, respectively), Reef 5 m (18.1% ± 4.7% and 7.6% ± 4.1%, respectively), and <31 °C marine lakes (18% ± 1% and 0.5% ± 0.7%, respectively). Reef 10 m and Reef 5 m had a significantly higher live benthic coverage of CCA (8.2% ± 4.4% and 9.4% ± 4.1%, respectively) compared to 31–32 °C (0.7% ± 0.9%) and >32 °C marine lakes (0.5% ± 1.2%). Hard coral was the most dominant group observed in reef sites, followed by turf algae and BCM. The most dominant group in <31 °C marine lakes was also hard coral, followed by CCA and turf algae. The coverage percent of Bivalvia from Reef 10 m (0.2% ± 0.4%) and Reef 5 m (0.2% ± 0.3%) was significantly lower than 31–32 °C (15.7% ± 13.9%)

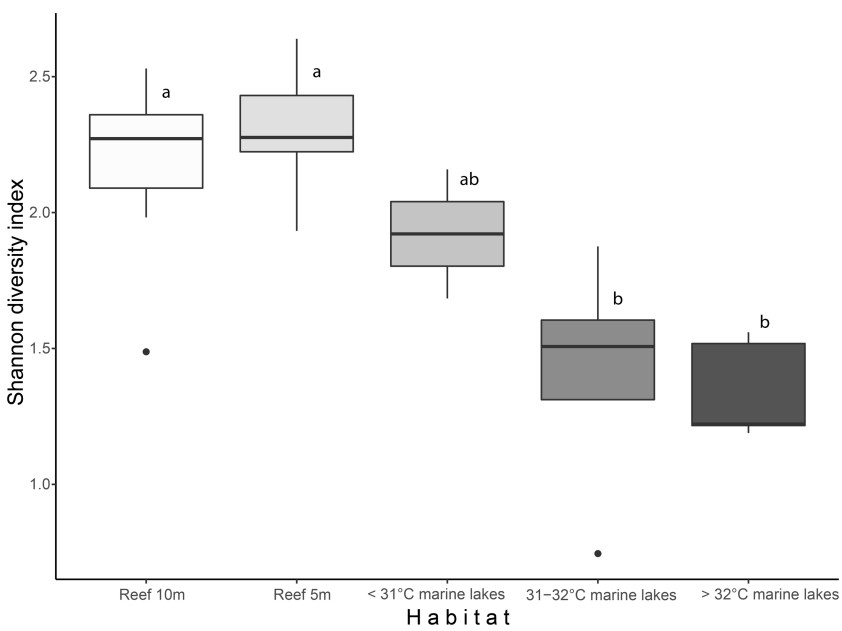

**Figure 2** **Boxplot of Shannon diversity index based on finer-scale benthic groups among habitats.** Different letters indicate significant differences (Kruskal–Wallis chi-squared $= 24.541$, $df = 4$, $p < 0.001$).

and >32 °C marine lakes (9% ± 6%). There were no significant differences in the cover of turfalgae, sponges, macroalgae, BCMs, 'other invertebrates', and substrate. The highest coverage was detected for turfalgae in Reef 10 m (15.7% ± 4.8%), sponges in >32 °C marine lakes (13.8% ± 7.1%), macroalgae in 31–32 °C marine lakes (21.9% ± 18.2%), BCM in 31–32 °C marine lakes (15.5% ± 12.6%), 'other invertebrates' in >32 °C marine lakes (12% ± 9.1%), and substrate in >32 °C marine lakes (24% ± 16.1%).

The cover of macroalgae, BCMs, Bivalvia, 'other invertebrates', and substrate in >32 °C and 31–32 °C marine lakes was relatively higher and had larger variation than <31 °C marine lakes, Reef 10 m, and Reef 5 m (Figs. 3 and 4 and Table S2). The bivalves found in the marine lakes predominantly comprised of mussels of the species *Brachidontes*. In the >32 °C marine lakes, no macroalgae detected from Papua 22 and Papua 7, but there was high coverage of *Caulerpa* in Papua 21 (33.8%). Similarly, in the 31–32 °C marine lakes, no macroalgae were detected in Papua 18, but there was a high coverage of *Halimeda* in Papua 2 (25.9%) and *Cladophora* in Papua 5 (35.9%). The most abundant macroalgae in reefs were *Halimeda* followed by *Padina* and *Sargassum*. The distribution of filter feeding organisms such as the mussel *Brachidontes*, Polychaetes, and Ascidians had a higher abundance in 31–32 °C and >32 °C marine lakes than in coral reefs. Whereas the abundance of those in marine lakes <31 °C was relatively similar to reefs and 31–32 °C and >32 °C marine lakes.

## Benthic community assemblage

There was a significant difference in benthic community composition across all the habitats (Adonis: $r^2 = 0.26$, $p < 0.001$). Specifically, there were significant differences among >32 °C marine lakes and Reef 5 m (pairwise adonis, $r^2 = 0.35$, $p = 0.01$), >32 °C marine lakes

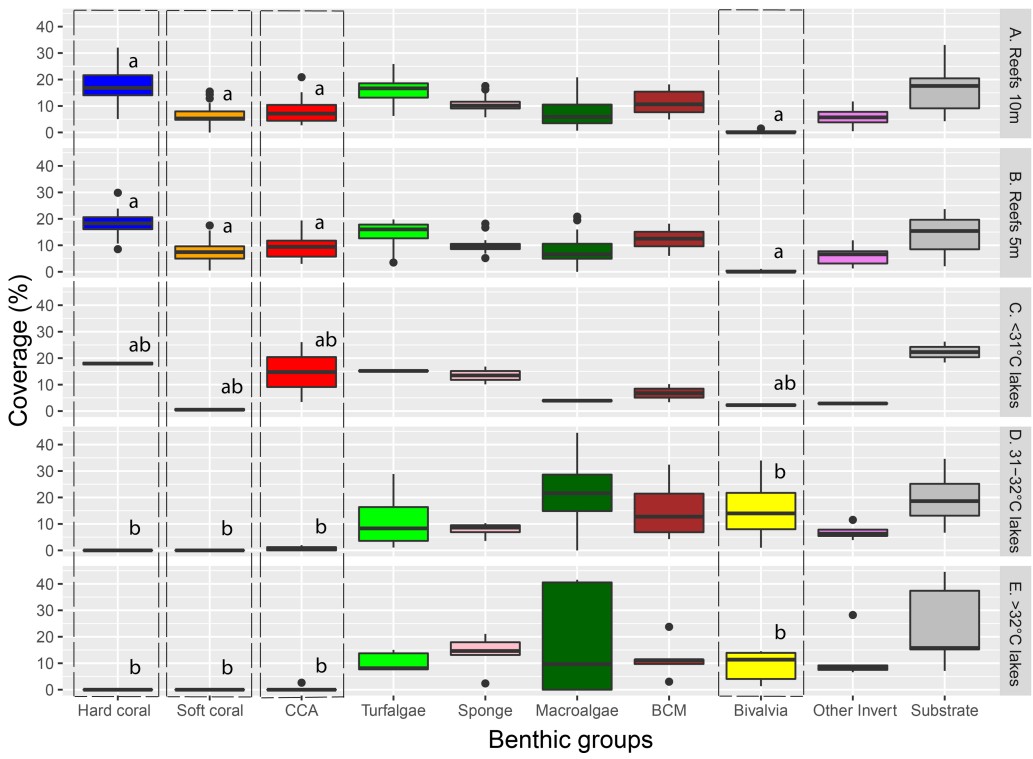

**Figure 3 The coverage of major benthic groups from each habitat category.** Different letters on Hard coral, Soft coral, CCA, and Bivalvia indicate significant differences across habitats (vertical line). Note: CCA, Crustose Coralline Algae; BCM, Benthic Cyanobacterial Mats.

and Reef 10 m (pairwise adonis, $r^2 = 0.34$, $p = 0.01$), 31–32 °C marine lakes and Reef 5 m (pairwise adonis, $r^2 = 0.34$, $p = 0.01$), and 31–32 °C marine lakes and Reef 10 m (pairwise adonis, $r^2 = 0.33$, $p = 0.01$). There were no significant differences in community composition among Reef 5 m and Reef 10 m, nor in comparisons among the reefs and <31 °C marine lakes.

NMDS ordination methods were used to show the clustering of benthic group composition (Fig. 5), where reef sites clustered together (Fig. 5A). The <31 °C marine lakes clustered towards the reef sites and were characterized by the presence of *Millepora*, mushroom coral, encrusting coral, excavating sponges, and CCA. The other marine lakes (31–32 °C and >32 °C marine lakes) did not cluster together. Finer-scale benthic groups of mussel, other BCM, and ball sponges were primarily found in marine lakes sites, whereas benthic groups of mushroom coral, CCA, branching coral, encrusting coral, massive coral, and other soft coral were most commonly found in reef sites, where they were abundant. Geographic location UTM_Easting ($r^2 = 0.3$, $p = 0.002$) and UTM_Northing ($r^2 = 0.41$, $p = 0.001$) influenced the composition of benthic communities in reef sites but depth ($r^2 = 0.1$, $p = 0.1$) and human activity ($r^2 = 0.06$, $p = 0.25$) did not (Fig. 5B). Degree of connection ($r^2 = 0.62$, $p = 0.026$), serving as a proxy for temperature, turbidity, and salinity, influenced benthic community composition in marine lakes, but geographic

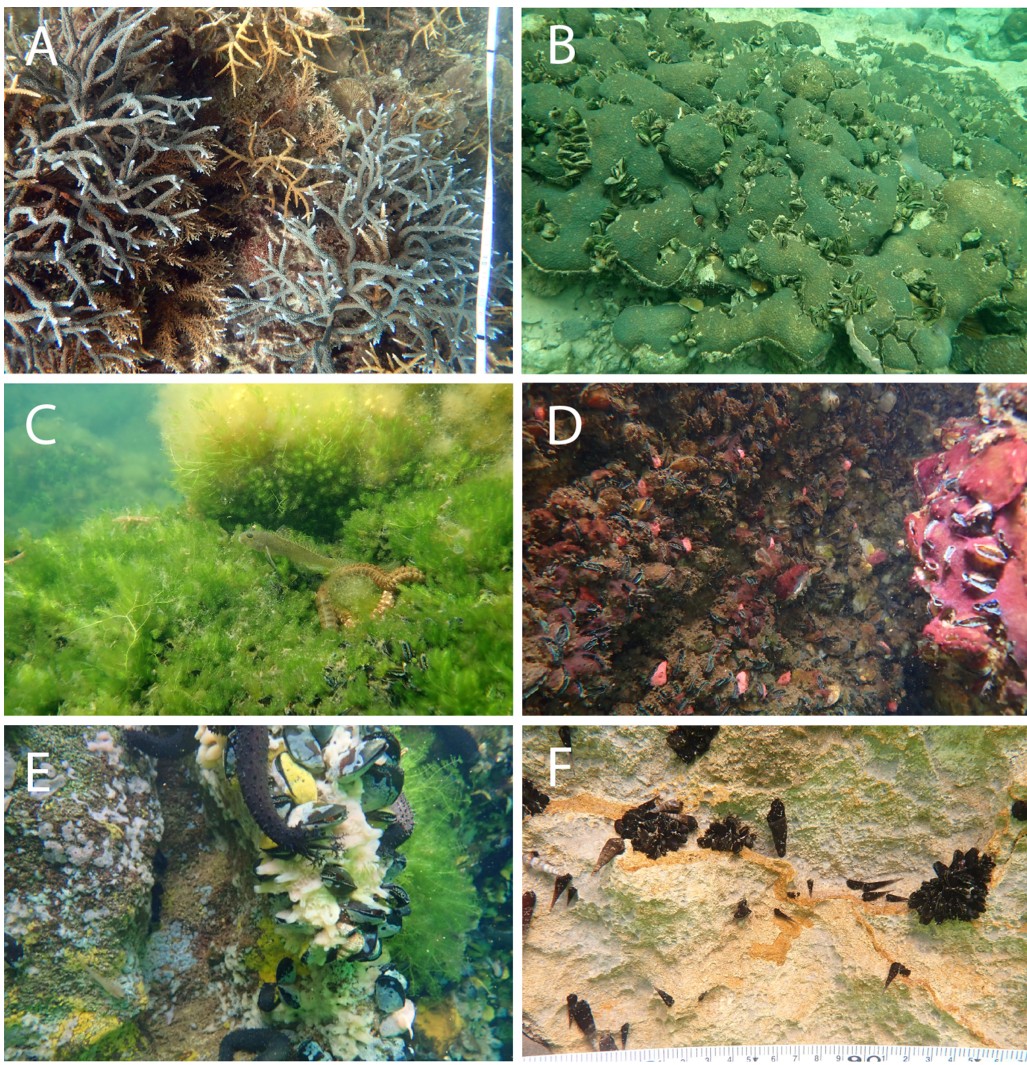

**Figure 4** Representative of benthic appearance from (A) Reef with branching *Acropora*, (B) <31 °C marine lake with massive coral, (C and D) 31–32 °C marine lake, and (E and F) >32 °C marine lake. Photo A (credit: Christiaan de Leeuw) and photos B–F (credit: Ludi P. Aji) were taken during data collection.

location UTM_Easting ($r^2 = 0.06$, $p = 0.755$) and UTM_Northing ($r^2 = 0.37$, $p = 0.16$) did not (Fig. 5C).

The *envfit* analysis showed that habitat was an important driver of benthic community assemblages ($r^2 = 0.56$, $p < 0.001$). The indicator finer-scale benthic group analysis ($p < 0.05$) identified three groups (other gastropoda, mussels, and other BCM) associated with >32 °C marine lakes; three groups (macroalgae *Cladophora*, mussels, and other BCM) associated with 31–32 °C marine lakes; seven groups (massive coral, submassive coral, encrusting coral, *Millepora*, CCA, excavating sponge, and digitate sponge) associated with <31 °C marine lakes; and 13 groups (branching *Acropora*, branching coral, foliose coral,
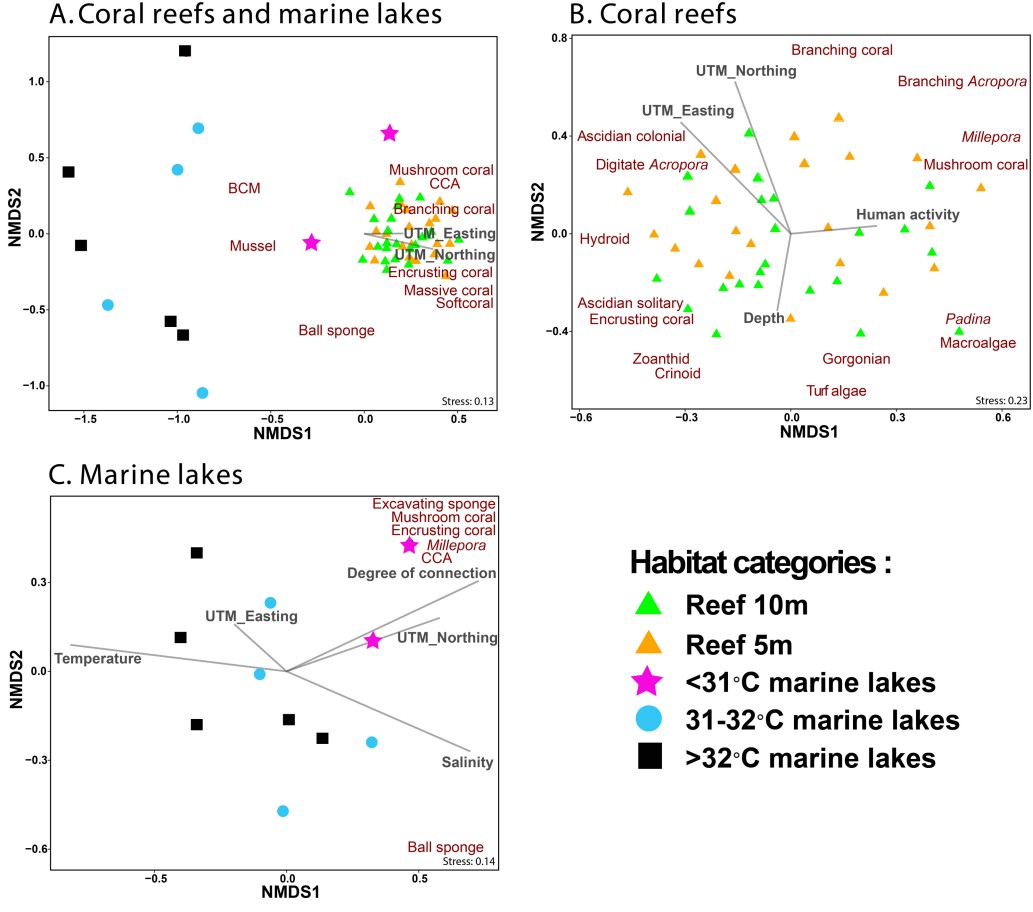

**Figure 5** **Non-metric Multidimensional Scaling (NMDS) ordination plot based on Bray–Curtis distances of benthic groups (A) from all marine lakes and reefs combined, (B) reefs, and (C) 11 marine lakes.** Benthic groups that significantly ($p < 0.001$) influenced the community assemblage are shown. The arrows represent environmental (temperature and salinity), physical characteristics (degree of connection and depth), geographic location (UTM_Easting and UTM_Northing), and human activity. Note: CCA, Crustose Coralline Algae; BCM, Benthic Cyanobacterial Mats.

massive coral, submassive coral, encrusting coral, mushroom coral, CCA, Gorgonian, other softcoral, Crinoid, Ascidian colonial, and Hydroid) associated with reef sites.

Mantel tests revealed significant correlations between benthic group composition (Bray–Curtis dissimilarity) and geographic distance. Specifically, a significant correlation was found when combining marine lakes and reefs from Misool ($r = 0.15$, $p = 0.013$), as well as when only including the reefs from Misool and Dampier strait ($r = 0.21$, $p = 0.011$). However, when we only included marine lakes, no significant correlation was observed between benthic communities and geographic distance ($r = 0.03$, $p = 0.38$). These results indicated that community assemblages of reefs were similar to other close reef sites, but this was not the case among marine lakes. However, the data showed a lot of spread and the correlation coefficient (*r-value*) was low, indicating a weak correlation between the variables (Fig. S4).

## DISCUSSION

Tropical coastal ecosystems are changing due to both global and local pressures (*Pandolfi et al., 2005*; *de Bakker et al., 2016*; *de Bakker et al., 2017*; *Teichberg et al., 2018*; *Xiao et al., 2022*). A key research question is how tropical coastal ecosystems could transition when subjected to continuous pressure from increased temperature and land-induced turbidity. This study of benthic communities across coral reef sites and marine lakes in three temperature categories (<31 °C, 31–32 °C, >32 °C) found that elevated seawater temperatures led to a decrease in biodiversity. Strikingly, the marine lakes with temperatures of 31 °C, despite the turbid conditions, had a similar relative coverage of hard corals, CCA, and macroalgae to the coral reef ecosystem. In marine lakes with temperatures higher than 31 °C, composition shifted towards the dominance of other benthic groups, such as macroalgae, BCM, and/or filter feeders, whereas hard corals and CCA were no longer present. The variation in the assemblage of benthic communities in the reef sites was significantly influenced by geographic distance between the sites, while temperature, salinity, and the degree of connection influenced the structure of benthic communities in marine lakes. The findings of the current study are further discussed below.

### Variation in composition among coral reef sites

The variation in the composition of assemblages in the reefs sites observed in this study can be explained primarily by geographic distance between the sites, and reef depth did not appear to influence community composition. These results are consistent with other studies that show geographic location to be a strong predictor of variation in benthic community composition in reefs (*McClanahan et al., 2014*; *Rattray et al., 2016*; *Yang et al., 2018*; *Obura et al., 2022*). Sites that are in closer proximity likely have similar (a)biotic and environmental conditions leading to similar relative coverage of benthic groups (*Rattray et al., 2016*; *Yang et al., 2018*). Contrary to the results of the present study, community composition has frequently been reported to be significantly different at different depths (*Cleary et al., 2005*; *Williams et al., 2013*; *Roberts et al., 2015*; *de Bakker et al., 2016*; *de Bakker et al., 2017*; *Cartwright et al., 2023*). However, most of these previous studies were conducted at varying depths from shallow water (∼3 m) to 40 m, differing from the 5 m and 10 m constant depths per site measured in the present study. It is possible that the physicochemical conditions at 5 m and 10 m are more similar within sites which support the turnover of larval benthic organisms (*Torruco, Gonzalez & Ordaz, 2003*; *Maldonado, 2006*; *Rattray et al., 2016*; *Yang et al., 2018*). Benthic larval propagules rely on water current for dispersal and are more likely to settle in the same site at a different depth than they are to settle at the same depth but in a farther site. The influence of depth on benthic community composition across reef sites could likely be detected by differences in light transparency, hydrodynamic energy, and physicochemical conditions if measured at deeper sites than the present study (more than 10 m).

The variation in the community structure in the coral reefs in Raja Ampat was not explained by the distance to human activity. These findings differ from previous studies in Indonesia, specifically those in the Thousand Islands (Jakarta Bay; *van der Meij, Moolenbeek & Hoeksema, 2009*; *Cleary et al., 2014*; *Baum et al., 2015*), and Spermonde Archipelago

(*Cleary et al., 2005*; *Becking et al., 2006*; *Teichberg et al., 2018*), which found that reefs closer to the mainland as the source of anthropogenic activity and eutrophication (decreasing water quality) caused nutrient enrichment which lead to the decreased biodiversity and coral coverage. Raja Ampat differs in two ways: (1) the area does not have large and highly-populated cities such as Makassar and Jakarta, which are significant sources of pollution (*Heery et al., 2018*), and (2) Raja Ampat has a geomorphology of multiple small islands intersected by profound depths and strong currents (*Sapin et al., 2009*; *Verstappen, 2010*), which is fundamentally different from the relatively shallow coastal shelves in the Thousand Islands and the Spermonde Archipelago, with clear land-sea cross-shelf gradients. The level of pollution caused by the types of human activities in Raja Ampat (pearl farms, villages, homestays, resorts) are possibly lower than seen in other studies (*Cleary et al., 2005*; *Baum et al., 2015*; *Teichberg et al., 2018*) and their effects may be further diluted by the strong currents and deep water close to the coasts of the islands. As currents were not considered in this study, and only the distance from the reef transect to the source of human activity was measured, this study may be missing a potential signal of the anthropogenic impact on benthic community assemblage. Future studies could include these variables as well as a direct measurement of *in-situ* water quality measurements.

This study showed a relatively lower coral coverage (average = 18.1%; range = 5.1–32.1%) than other reef monitoring in Indonesia. Raja Ampat is known for its relatively pristine coral reef sites with many sites lauded by scuba divers for high coral diversity (*Mangubhai et al., 2012*; *Yuanike et al., 2019*; *Maas et al., 2020*; *Purwanto et al., 2021*). However, the current study did not specifically target the best dive-sites, nor the best protected sites within the Marine Protected Areas, but rather systematically chose sites that were close to the marine lake sites for comparison and targeted similar coastal settings to avoid confounding factors. To protect coral reef ecosystems, the Indonesian government has implemented the Coral Reef Rehabilitation and Management Program - Coral Triangle Initiative (Coremap-CTI), which performs coral reef monitoring across Indonesian reefs (*Hadi et al., 2019*). This initiative categorizes the hard coral coverage in coastal reefs as: poor (<25%), fair (25%–50%), good (50%–75%), or excellent (>75%). Monitoring results in Raja Ampat found the average (min-max) percentage of live hard coral coverage was: 28% (10.1–44.9%) in Salawati Batanta (*Rondonuwu et al., 2019*), 26% (5.8–47.2%) in Wayag (*Abrar et al., 2015*), 38% (17.8–68.9%) in West Waigeo (*Rizqi et al., 2019*), 64% (44.2–86.8%) in Dampier straits (*Yuanike et al., 2019*), 31% (18.1–61.3% in year 2019) in North Raja Ampat, and 39% (34.3–46.3%) in South Raja Ampat (*Purwanto et al., 2021*). These monitoring results indicate that overall coral coverage in Raja Ampat's reefs, which are mostly located in marine protected areas, can be categorized as fair to good (*Hadi et al., 2019*; *Pakiding et al., 2019*; *Purwanto et al., 2021*). Survey results of 1,153 locations across Indonesian reefs surveyed through the Coremap-CTI program in 2019 using the same methods as the present study (underwater photographic transect) resulted in 390 (33.8%) reef sites classified as having poor coral coverage, 431 (37.4%) having fair coral coverage, 258 (22.4%) having good coral coverage, and 74 (6.4%) reef sites having excellent coral coverage (*Hadi et al., 2019*). Thus, continued monitoring in a diversity of habitats is essential to assess the general health condition of the coral reefs in Raja Ampat and to

provide a better understanding of the primary factors influencing the condition of these coral reefs.

## Coral communities in marine lakes

The marine lakes with temperatures of <31 °C contained a cover of hard coral (17.4–18.8%), CCA (3.5–26.3%), and turf algae (15–15.5%) that was similar to the coral reef sites. Using the conceptual framework introduced by *Schoepf et al. (2023)* to define marginal and extreme coral communities, marine lakes in this study appeared to harbour extreme coral communities, as they can survive under the chronic extreme environmental conditions of higher temperatures, land-based sedimentation and in some cases with lower salinity levels. It is important to note that the marine lakes in the current study that contained corals had an average temperature of 30.4 °C, which is markedly higher than the average temperatures recorded in the reefs of Raja Ampat of 27.9 °C (averaged from 2009–2011; *Purwanto et al., 2012*, 27.3 °C (year 2019; *Patty et al., 2020*), and 28.5 °C (*Nugraha et al., 2018*). While the diversity of the major benthic groups was similar among the reef sites and the marine lakes of <31 °C, the diversity of coral growth forms was lower in the marine lakes compared to the reefs. The corals in the marine lakes were predominantly massive, submassive, and encrusting, which are known to be more common in coral communities under turbid and extreme environmental conditions (*Santodomingo, Renema & Johnson, 2016*; *Camp et al., 2018*; *Evans et al., 2020*; *Schoepf et al., 2023*). In contrast, branching *Acropora,* branching coral, foliose coral and soft coral were primarily found in the coral reef sites and were rare in marine lakes. These groups are generally not resilient to high sedimentation and limited current flow (*Loya et al., 2001*; *De'ath & Fabricius, 2010*; *Qin et al., 2020*). The *Millepora* hydrocoral was abundant in marine lakes and seemed to be adapted to the marine lake environment, as they are fast-growing and can recover rapidly from disturbance (*Lewis, 2006*; *Brown & Edmunds, 2013*). Further study of coral species diversity is required to determine which species can tolerate marine lake environments.

## Decline in diversity and shift in dominant benthic groups

The diversity in the marine lakes was lower as temperatures rose. Corals and CCA (encrusting and calcifying red algae) were not found in the marine lakes with temperatures of more than 31 °C. High temperatures, suspended particulate matter, and dissolved inorganic nutrients can inhibit fertilization, fecundity, and larval settlement of coral (*Fabricius, 2005*; *Haas et al., 2016*). CCA, which contribute to building and solidifying the reef structure (*Lewis, Kennedy & Diaz-Pulido, 2017*; *Weiss & Martindale, 2017*; *Chen et al., 2020*) and offer settlement cues for coral larvae (*Whitman et al., 2020*; *Deinhart, Mills & Schils, 2022*; *Abdul Wahab et al., 2023*), have been found to be sensitive to thermal stress (*Webster et al., 2011*; *Vásquez-Elizondo & Enríquez, 2016*). A positive correlation was observed between increased CCA fragments and an increase in species, type, and coverage of hard corals (*Weiss & Martindale, 2017*; *Whitman et al., 2020*; *Abdul Wahab et al., 2023*). This suggests that an increased availability of CCA in the ecosystem likely increases the diversity and coverage of hard corals. A decrease in biodiversity might be accompanied by a lack of functional redundancy (*Rosenfeld, 2002*; *Biggs et al., 2020*) and have a severe

effect on interactions across trophic levels, leading to unexpected changes in community dynamics (*Palumbi, McLeod & Grünbaum, 2008*).

The composition of assemblages in the marine lakes varied greatly in this study and was mostly explained by water temperature. In the marine lakes with higher temperatures (31–32 °C and >32 °C), where there was no coral dominance, the systems were dominated by macroalgae (both fleshy and calcareous), BCMs, sponges, filter feeders, or a combination of these groups.

Macroalgae are generally associated with shifts in dominance of coral reefs that are under stress from high temperatures, nutrient pollution, and overfishing (*Vroom et al., 2006*; *Littler, Littler & Brooks, 2009*; *Fulton et al., 2019*; *Reverter et al., 2020*; *Crisp, Tebbett & Bellwood, 2022*). Macroalgae coverage in the current study, however, varied widely among marine lakes with higher temperatures (>31 °C). In particular, in the marine lakes with the highest temperatures, macroalgae were absent or only present with low coverage percentages. This variation in dominance could be due to differences in the macroalgae species present in different marine lakes. Macroalgae can be classified as fleshy, filamentous, or calcareous taxa, and different macroalgae species have different responses to environmental variables (*Vroom et al., 2006*; *Diaz-Pulido et al., 2009*; *Cannon et al., 2023*). The fleshy macroalgae *Caulerpa* was dominant in Papua 21, the filamentous macroalgae *Cladophora* was dominant in Papua 5, and the calcareous macroalgae *Halimeda* was dominant in Papua 2. Calcareous macroalgae, such as *Halimeda* and *Peyssonnelia,* which are less harmful for hard corals, were found in <31 °C marine lakes where hard coral is present. It is possible that when the temperatures rise beyond a certain level, the dominance of macroalgae shift to other benthic groups that are more stress tolerant, such as BCMs (*Tebbett et al., 2022a*) and filter feeders (*Huhn, 2016*). Conversely, turf algae, which is a heterogenous consortium of fleshy, short filamentous algae, juvenile macroalgae, and cyanobacteria (*Adey & Steneck, 1985*; *Bender, Diaz-Pulido & Dove, 2014*; *Harris, Lewis & Smith, 2015*; *Tebbett et al., 2022b*), had similar coverage across all the reef sites and marine lakes in this study. Turf algae is opportunistic (*Bender, Diaz-Pulido & Dove, 2014*; *Connell, Foster & Airoldi, 2014*) and can rapidly occupy open space (*Harris, Lewis & Smith, 2015*; *Tebbett et al., 2022b*), particularly when the conditions are not optimal for corals (*Fong & Paul, 2011*). This suggests that turf algae may dominate space under various stress conditions and disturbances. Both macroalgae and turf algae can inhibit coral recruitment, coral reproduction, overgrow/shade coral, disrupt the coral microbiome, and increase pathogenic bacteria (*Fong & Paul, 2011*; *Wild, Jantzen & Kremb, 2014*; *Harris, Lewis & Smith, 2015*; *Haas et al., 2016*; *Fulton et al., 2019*), allowing them to outcompete coral in reef ecosystems.

BCMs, which are a consortium of microbes often dominated by cyanobacteria (*Charpy et al., 2012*; *Huisman et al., 2018*), were present across all coral reefs and marine lakes in this study, ranging from 3% to 33% coverage. The highest coverage was found in 31–32 °C marine lake Papua 18, followed by >32 °C marine lake Papua 7. Red-brown mat was the most common BCM type found in both marine lakes and coral reefs. This is similar to the finding of *Stuij et al. (2023)* that red-brown mat was one of the most observed BCM on coral reefs of Koh Tao in the Gulf of Thailand. BCMs are becoming emerging players in

coral communities as several studies have documented an increased abundance of BCMs in response to elevated temperatures, nutrient availability, and reduced water quality (*de Bakker et al., 2017*; *Ford et al., 2017*; *Ford et al., 2018*; *Ford et al., 2021*; *Tebbett et al., 2022a*; *Stuij et al., 2023*). In large quantities, BCMs could negatively affect ecosystems as they can cause local anoxia, contain pathogens, smother benthos, and prevent larvae from settling (*Nagle & Paul, 1998*; *Charpy et al., 2012*; *Ford et al., 2018*; *Ford et al., 2021*). In a study of an inshore reef system in Fiji, *Ford et al. (2021)* found that BCMs are herbivory-resistant primary producers, as the fish that typically eat algae or corals showed little interest in consuming BCMs. There are less fish in marine lakes than in coral reef ecosystems, with most fish being small omnivores (pers. obs.; *Becking et al., 2011*). Because of this, marine lakes could be considered a proxy for an 'overfished' system, where BCMs and algae have little top-down control. Mesocosm experiments under future climate conditions found that the systems shifted with BCMs having more biomass under high temperature conditions (*Ullah et al., 2018*; *Nagelkerken et al., 2020*). With increasing temperatures and external organic matter input in tropical coastal ecosystems, heat-tolerant BCMs will likely be able to rapidly transform external organic matter into biomass, leading to rapid growth and possible dominance in benthic coverage (*Charpy et al., 2012*; *Brocke et al., 2015*).

Sponges have been hypothesized to be a possible emerging benthic group in the transformation of coral reef ecosystems as they have a competitive advantage to corals through temperature tolerance, chemical defenses, symbiotic relationships, adaptation to variable conditions, and mixotrophic and sponge-loop mechanisms (*Bell et al., 2018*; *Pawlik, Loh & McMurray, 2018*; *Zuschin, Hohenegger & Steininger, 2001*). Whilst we observed sponges across all of the marine lakes and the coverage was high in some, we did not find any evidence that sponge cover varied significantly with differences in temperature. As such, our work does not show that sponges necessarily replace corals as environmental stress increases (for example through increases in temperature or eutrophication). Further analysis of the variation in sponge species would provide more insight into how sponge species communities adapt in response to the environmental change.

We found that filter feeders, such as bivalves and polychaetes, dominate high temperature and turbid marine lakes. Our results align with various studies that have shown that degraded reefs under eutrophic settings can have a high abundance of coral-associated bivalves and polychaetes (*Chazottes et al., 2002*; *Hutchings & Peyrot-Clausade, 2002*; *Mohammed & Yassien, 2008*; *Hoeksema et al., 2022*). Tubeworms have been seen to form associations with hard corals and sponges, exhibiting increased abundance in environments characterized by turbidity, increased exposure, and eutrophic and harsh conditions (*Ben-Eliahu, Safriel & Ben-Tuvia, 1988*; *Schwindt, Bortolus & Iribarne, 2001*; *Hutchings & Peyrot-Clausade, 2002*; *Hoeksema et al., 2022*). As they are bioeroding, these filter feeders weaken the structural strength of the reefs (*Schwindt, Bortolus & Iribarne, 2001*; *Fabricius, 2011*), which can result in the development of abnormal growth forms that inflict damage on hard coral hosts, leading to an increase in dead hard coral patches that are utilized as substrate by other benthic groups such as sponges and algae (*Hoeksema et al., 2022*).

The mussel *Brachidontes* of the Mytilidae family was found in all marine lakes in this study and was highly abundant in marine lakes with temperatures above 31 °C, where

they attached themselves to the hard substrata or mangrove roots (*Becking et al., 2011*; *Aji et al., 2023*). Mussels are frequently seen to dominate the benthic biomass in marine lake ecosystems in response to turbidity and nutrient enrichment (*de Leeuw et al., 2020*; *Aji et al., 2023*). High turbidity limits light, negatively impacting benthic phototrophic species, but promoting heterotrophic filter feeders (*Fabricius, 2005*; *Fabricius, 2011*). High phytoplankton and particulate organic matter load in eutrophic aquatic environments may increase the survival of bivalves (*Nakamura & Kerciku, 2000*). The mussel *Brachidontes*, as a primary consumer, can survive under large fluctuations in salinity and temperature conditions that are usually lethal for the majority of other benthic groups (*Safriel & Sasson-Frostig, 1988*; *Sarà et al., 2008*; *Astudillo, Bonebrake & Leung, 2017*). *Huhn (2016)* showed that bivalve mussels are more heat resistant and can tolerate temperatures up to 39 °C. This suggests that *Brachidontes* may be able to physiologically regulate and maintain feeding and food acquisition in a broad range of environmental conditions. Therefore, filter feeder organisms could be a potential opportunistic benthic group that is able to dominate tropical coastal ecosystems with high temperatures and turbidity.

## Marine lakes as possible model systems

In recent years there has been heightened interest in corals that live at the edge of their environmental limits as they can provide insights into how coral communities may survive and adapt to future changes to marine environments (*e.g.*, *Schoepf et al., 2023*). The current average temperature of reefs in Raja Ampat is around 29 °C (*Purwanto et al., 2012*; *Nugraha et al., 2018*; *Patty et al., 2020*). Marine lakes, however, naturally represent possible climate change scenarios predicted by the IPCC under the worst case emissions model. For example, marine lakes Papua 7 and Papua 22 have temperatures >35 °C, as well as an anoxic layer in the water column. Under the most severe IPCC scenario (RCP8.5), global sea surface temperature is projected to rise by approximately 2.58 °C by the year 2100 (*IPCC, 2014*). This increase in temperature, coupled with melting ice caps and increased rain, may lead to reduced oceanic salinity (*IPCC, 2021*). Some evidence suggests that the Coral Triangle Area was more brackish during the Last Glacial Maximum (17–18 K years ago), with the last melting of the ice caps, than it is now (*Hoeksema, 2007*). Increase water temperatures, salinity changes, together with increased land-based pollution of nutrients from runoff of agricultural fertilizers and sediment, are expected to promote stratification in coastal waters, leading to more frequent hypoxic or anoxic events (*Gooday et al., 2009*; *Zhang et al., 2010*).

Marine lakes are not exact predictors for climate change effects in coastal systems, but are highly informative for four reasons: (1) marine lakes have high temperatures and turbidity on a sufficiently large spatial and temporal scale to integrate ecosystem processes; (2) marine lakes provide an understanding of interactions of major benthic groups under variable environments and during dominance of certain benthic groups; (3) in the Indo-Pacific geomorphology of the reefs, particularly in karstic systems such as West Papua, multitudes of basins with restricted waterflow are common; and (4) proxies and island systems, such as hydrothermal vents, have been useful indicators in the past (*Pichler et al., 2019*; *Schoepf et al., 2023*). Further research is needed on how BCMs, filter feeders,

and sponges, together with corals, macroalgae, and turf algae, will develop under changing conditions as the major groups that are dominating the marine lakes appear to be the same as those emerging in the reefs. The marine lake setting may provide an opportunity to understand and identify what bottom-up processes could lead to major shifts in the ecosystem.

## CONCLUSIONS

Studying the benthic communities in both coral reefs and marine lakes provides insights into potential changes in tropical coastal ecosystems when environmental variables and water quality change. The biodiversity of benthic communities decreased with increased seawater temperatures, with a shift from coral-algae dominated systems to those dominated by macroalgae, BCMs, filter feeders, sponges, or a combination of these groups. Our results suggest that beyond a certain temperature (>31 °C), tropical benthic communities may shift away from coral dominance, but new outcomes of assemblages can be highly distinct, with a possible varied dominance of macroalgae, benthic cyanobacterial mats, or filter feeders such as bivalves and tubeworms. Macroalgae were not present in the marine lakes with the highest temperatures (>35 °C), which were dominated by BCMs and filter feeders. A key question that needs further research is what a shift in dominance to BCMs and filter feeders may mean for the functioning of coastal communities, as this shift may lead to altered ecological function and affect energy flow from primary producers to higher consumers (*Goldenberg et al., 2017*; *Ford et al., 2018*; *Cissell & Mccoy, 2022*). In conclusion, while marine lakes do not directly translate to coastal reef systems, they can provide insights into the interactions of major benthic groups under variable environmental conditions and the bottom-up processes that could lead to shifts in dominance within tropical coastal marine ecosystems if hard corals are no longer dominant.

## ACKNOWLEDGEMENTS

We would like to thank Augy Syahailatua, Syafri, Ricardo Tapilatu, Purwanto, Ali Oherenan, Stephanie Martinez, Robin Olde Wolbers, Inez van Erp, Merethe, Anika, Esmee, Max Ammer, Andy Miners, UPTD BLUD, Universitas Papua, Conservation International Indonesia, Baseftin Foundation, Raja Ampat Research and Conservation Center, Papua Diving, and Misool Eco Resort. We would also like to thank the reviewers for their support in improving the manuscript.

### Funding

This work was financially supported by the Endowment Fund for Education (LPDP) Scholarship from the Ministry of Finance of the Republic of Indonesia for Ludi Parwadani Aji and the Dutch Research Council (NWO) project VI.Vidi.193.137 for Leontine Becking. The funders had no role in study design, data collection and analysis, decision to publish, or preparation of the manuscript.

## Grant Disclosures

The following grant information was disclosed by the authors:
Ministry of Finance of the Republic of Indonesia.
Dutch Research Council: VI.Vidi.193.137.

## Competing Interests

The authors declare there are no competing interests.

## Author Contributions

- Ludi Parwadani Aji conceived and designed the experiments, performed the experiments, analyzed the data, prepared figures and/or tables, authored or reviewed drafts of the article, and approved the final draft.
- Diede Louise Maas conceived and designed the experiments, analyzed the data, authored or reviewed drafts of the article, and approved the final draft.
- Agustin Capriati performed the experiments, authored or reviewed drafts of the article, and approved the final draft.
- Awaludinnoer Ahmad performed the experiments, authored or reviewed drafts of the article, and approved the final draft.
- Christiaan de Leeuw performed the experiments, authored or reviewed drafts of the article, and approved the final draft.
- Leontine Elisabeth Becking conceived and designed the experiments, authored or reviewed drafts of the article, and approved the final draft.

## Field Study Permissions

The following information was supplied relating to field study approvals (i.e., approving body and any reference numbers):

Field works were approved by Kementrian Riset, Teknologi, dan Pendidikan Tinggi (1353/FRP/E5/Dit.KI/III/2018), and Lembaga Ilmu Pengetahuan Indonesia (B-7/IPK.2/KS.01/I/2020)

## Data Availability

The raw measurements are available in the Supplementary File.

The GPS coordinate location of marine lakes in Raja Ampat is restricted to protect the vulnerable ecosystems. The location must be requested from Bayu Prayudha, bayu005@brin.go.id, from Research Center for Oceanography - BRIN. The requester should be a scientist from an academic research institute, must include a statement of purpose of the scientific use of the coordinate, and must sign an agreement not to share the coordinate to third parties. Requests will be evaluated and responded to within 1 month.

## Supplemental Information

Supplemental information for this article can be found online at http://dx.doi.org/10.7717/peerj.17132#supplemental-information.

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
