# Peer review of "Shifts in dominance of benthic communities along a gradient of water temperature and turbidity in tropical coastal ecosystems"

_PeerJ, doi:10.7717/peerj.17132_

## Round 0.1 · original submission · Minor Revisions

Dear Authors,
two reviewers revised the manuscript and both of them found the paper well-written, clear, and with interesting results. Nevertheless, they require some minor revisions to better clarify the work and improve it. Please provide us with a new revision version.
Sincerely
Federica Costantini

·

Basic reporting

no comment

Experimental design

please describe temperature data more clearly, see additional comments

Validity of the findings

While the validity of the findings are sound, I feel caution needs to be applied when comparing present marine lake environments to future coral reefs under climate change scenarios. While salinity and temperature of future coral reefs might be similar to current marine lakes, other environmental factors such as connectivity, currents and wave energy make these environments inherently different. The manuscript does acknowledge this to some extent.

Additional comments

line 167 – Can you describe the temperature data more clearly? i.e., how many times/ across what time period were the temperature measurements taken? Was it just one-off measurements or is this a mean temp? Also, were seasonal differences/ daily fluctuations in temp looked at? E.g., temperature variability across time is known to be a factor in coral resilience to heatwaves, (Safaie, Aryan, et al. "High frequency temperature variability reduces the risk of coral bleaching." Nature communications 9.1 (2018): 1671.) and therefore regime shifts/algal colonization.
line 186 – how was the distance between photos decided – random, systematic, haphazard etc?
line 364 – similar to what (each other or coral reefs)? Can you clarify?

Reviewer 2 ·

Basic reporting

The paper is very well written and easy to follow with all the correct sections and clear tables and figures. References to the literature are sufficient in number and appropriate for the content. There are no hypotheses, but the study states clear aims in the introduction which are largely achieved through the sampling and analyses undertaken.

Experimental design

This primary research study is suitable for publishing by PeerJ in terms of scope. The research aims are well defined, highly relevant in the context of global climate change and have genuine real-world applicability. A rationale for the study approach and identification of the knowledge gap that this research will fill is provided, but these are not entirely clear. Similarly, field and data analysis methods are mostly clear and rigorous but are at times missing details that hinder a full assessment of the methods used. Specifically:

Lines 144 & 145 – Expand and clarify why you consider it appropriate to use marine lakes to provide insights about open reef ecosystems, I would suggest inserting here the four points listed in the discussion (lines 613-620).

Lines 127-128 – It is not clear how the knowledge gap you are identifying differs from the studies referenced in the opening of this paragraph (did de Bakker et al., 2017, Giorgi et al., 2022 and Caballero-aragón et al., 2023 not investigate ecological responses of the community over several decades?).

Lines 160 to 164 – surveys in reefs and lakes took place in different seasons and two years apart (reef sites were surveyed in Apr/May 2018 and marine lakes were surveyed Jan/Feb 2020). Could the authors please comment on how they might expect these differences in timing to influence community composition in each ecosystem and whether this limits the validity of comparisons between them?
Line 164 – How were the temperature categories for assessing difference between lakes chosen? For instance, was this based on some literature evidence of breakpoints in community composition at these temperatures?

Lines 183-184 and 186-188 – The transect setup is not clear for either lakes or reefs. Was there only one transect in each lake (as indicated in line 181) or three (as indicated in line 183)? How many transects were there in total at each reef site, three, six, nine, some other amount? Please rephrase lines 186-188 to be clearer.

Lines 186 and 188 – Why were different sized photos used in the marine lakes versus reef sites?

Lines 252-253 – It is not clear to me what the geographic distance variable is measuring and why. Is it the distance from one site to the next closest site? If so I would edit this sentence to state this clearly, e.g. “The geographic distance variable referred to the distance between survey sites and was calculated as…”. Is this considered important because you would expect sites that are closer to be more similar in their community composition (if so, is this the analysis that is described in lines 348-349)?

Lines 254-255 – I do not understand what "geographic data" are and how they might differ to "geographic distance". In the previous sentence it is explained that distance was measured from the centre of each site, but here it is stated geographic data was measured from 5m depth. Can you please clarify.

Validity of the findings

Underlying data have been provided in a clear and succinct format. The statistical analyses are mostly sound to achieve the aims of the study and have been applied correctly, except for some specific instances which I have detailed comments for below. The conclusions the authors are trying to make are sometimes unclear and need more thorough explanation. Specifically:

Lines 189-190 – Given different surface areas were sampled in lakes versus reefs, were univariate community metrics such as richness and diversity standardised to a common area? What impact do the authors consider these differences in surface area might have had on these metrics?

Lines 208-210 – I think it would be useful to include geographic location and depth in the collinearity tests, given both of these factors would be expected to influence temperature. If these variables are included independently in regression analyses then there may be issues when fitting and interpreting the model.

Lines 274-277 – Do these statistics come from tests where all reef data (5 and 10 m depths) are pooled and all lakes data (all three temperature categories) are pooled? If so, I would explain this more clearly and remove the reference to Figures 2 and S1 as this is not what is shown in the figures (which demonstrate that there is no difference between the <31 oC lakes and reefs).

Lines 333-334 – It is not appropriate to include degree of connection, temperature and salinity in regression analyses as these are all collinear (as stated in Lines 269-271), pick one (e.g., degree of connection) and explain that this also acts as a proxy for the others (i.e., temperature and salinity).

Lines 315-316 – Do these statistics come from tests where data for the reefs and lakes were pooled into two categories? If so explain this more clearly.

Lines 348-349 – I do not find the analyses being described here very clear. Are you looking at correlations between distance matrices of different sites and relating the results to how far apart the sites are? This is likely a continued confusion about what is meant by "geographic distance", better explanation of this in the methods should help with comprehension of these results.

Figure 5 – I do not think it makes sense to include depth on the coral reefs nMDS, given other statistical analyses performed in the paper that there is no significant difference between the 5 and 10 m sites.

Lines 547-558 – This paragraph is not very impactful as it is not clear what point you are trying to make. Are you trying to convey the theory that sponges are expected to replace corals as environmental stress increases (for example through increases in temperature), and that whilst you observed sponges across all of your marine lakes and the coverage was sometimes high, you did not find any evidence that sponge cover varied significantly with differences in temperature?

Lines 560-575 – It is not clear how this paragraph relates to your findings, it reads like textbook information without any reference to the present study. Either remove this paragraph or incorporate some evidence from your results that either support or challenge the information being presented.

Additional comments

Introduction
Lines 105-106 – This statement is inaccurate, there is extensive research and knowledge of how stressors like sediment, nutrients, organic matter etc. affect soft sediment benthic communities. I suggest specifying that you mean tropical benthic reef communities, perhaps?

Results
Lines 263-264 – It is stated that Table 1 contains environmental characteristics for the marine lakes and reef however there are no environmental data for the reef sites. Can you amend this sentence and explain why this is, including a statement about what impact this has on your ability to make comparisons between reefs and lakes? Also suggest editing Table 1 to remove reefs if there are no data to present.

Lines 277-281 – The written explanation of which reefs differ significantly from which lakes is hard to follow and repetitive of the information in the figures, suggest deleting.

Lines 308-309 – This sentence is a bit hard to follow, suggest rephrasing to: "The distribution of filter feeding organisms such as the mussel Brachidontes, Polychaetes and Ascidians were found in..."
Lines 311-312 – How did the abundance of filter feeders in lakes <31 oC compare to the other habitats?

Discussion
Lines 364-365 – It is not surprising that lakes of a similar temperature and turbidity would have similar benthic assemblages. Do you mean “despite the differences in turbidity between them”? Clarify this statement.

Lines 382-384 – Is it also true that sites in close proximity are likely to have similar relative coverage of benthic groups because they are likely fed by the same adult source populations?

Lines 415-419 – Perhaps another suggestion would be to categorise human activities and treat these as independent variables for analyses, as the different types of activity are likely to impact benthic communities in different ways.

Line 470-471 – Is it not possible to ascertain from your study whether the corals within the low temperature lakes are the more resilient types?

Lines 520-521 – “Thus, the >31 °C marine lakes, which are harsh environments for coral, are likely to be occupied by macroalgae and/or turf algae”. This kind of statement reads like a hypothesis and should be made in the introduction rather than the discussion, your results have already shown this to be true. Rephrase to “…harsh environments for coral, were occupied by…”.

Lines 603-604 – It would be helpful to remind the reader of the current average temperatures of reefs in Raja Ampat that are listed in lines 454-458, to help understand how the temperatures of the marine lakes relate to predicted future reef temperatures.

Lines 622-623 – “…the major groups that are dominating the marine lakes appear to be the same as those emerging in the reefs.” This is an interesting point worth exploring more in the discussion, which of your results demonstrate this pattern?

Lines 641-643 – Expand on why you think a focus on the degree of determinism versus stochasticity in community shifts would be worthwhile future research.

Tables & Figures
Figure S1 – I am not sure if I am misinterpreting the letters on the figure but it seems as though the richness of lakes <31 oC is not significantly different to 31-32 oC and >32 oC lakes, which I find hard to believe.

Figure 3 – It appears that the coverage of soft corals in <31 oC lakes is almost identical to that in 31-32 oC lakes and >32 oC lakes, yet only <31 oC lakes are not significantly different to the reef habitats?

Table 1 – Provide more detail to the caption to explain the temperature and salinity measures e.g., are these means or medians? Were multiple measurements taken over time, or were several locations (different depths?) sampled within each lake?

Annotated reviews are not available for download in order to protect the identity of reviewers who chose to remain anonymous.

---

## Round 0.2 · accepted · Accept

Dear Authors, I am very happy to inform you that the paper you have submitted is accepted for publication. I read the manuscript and the rebuttal and I found that you did a good job in replying to the reviewers and addressing all their comments.